# Biocompatible Cantilevers for Mechanical Characterization of Zebrafish Embryos using Image Analysis

**DOI:** 10.3390/s19071506

**Published:** 2019-03-28

**Authors:** Yuji Tomizawa, Krishna Dixit, David Daggett, Kazunori Hoshino

**Affiliations:** 1Department of Biomedical Engineering, University of Connecticut, Storrs, CT 06269, USA; yuji.tomizawa@uconn.edu (Y.T.); krishna.dixit@uconn.edu (K.D.); 2Department of Molecular & Cell Biology, University of Connecticut, Storrs, CT 06269, USA; david.daggett@uconn.edu

**Keywords:** stiffness analysis, force sensor, zebrafish embryo, biosolid mechanics, soft lithography

## Abstract

We have developed a force sensing system to continuously evaluate the mechanical elasticity of micrometer-scale (a few hundred micrometers to a millimeter) live tissues. The sensing is achieved by measuring the deflection of force sensitive cantilevers through microscopic image analysis, which does not require electrical strain gauges. Cantilevers made of biocompatible polydimethylsiloxane (PDMS) were actuated by a piezoelectric actuator and functioned as a pair of chopsticks to measure the stiffness of the specimen. The dimensions of the cantilevers were easily adjusted to match the size, range, and stiffness of the zebrafish samples. In this paper, we demonstrated the versatility of this technique by measuring the mechanical elasticity of zebrafish embryos at different stages of development. The stiffness of zebrafish embryos was measured once per hour for 9 h. From the experimental results, we successfully quantified the stiffness change of zebrafish embryos during embryonic development.

## 1. Introduction

The study of three-dimensional micro-mesoscale (100 µm–1 mm) tissues, such as multicellular spheroids [1,2,3,4,5,6], tissue organoids [7,8,9,10,11], and animal embryos [12,13,14], is a topic of recent interest. The study of biomechanics in such tissues can provide a deeper understanding of the differentiation, migration, and proliferation of cells. Commercially available atomic force microscopy (AFM) has already shown success in the mechanical characterization of single cells [15,16] and flat tissue sections [17]. However, the sensing cantilever of the AFM only operates in a limited degree of motion and is not suitable for the study of fully three-dimensional tissues at the micro-mesoscale. Several microfabricated silicon cantilevers integrated with piezoresistive strain gauges have demonstrated the versatility needed for force sensing [18,19,20,21] and micromanipulation [22,23]. However, the lithography-based fabrication process required to make the force sensitive cantilevers is expensive, limiting their use in biomedical applications where low-cost disposable components are desired. 

Here, we propose a force sensor system based upon microtweezers, modified from our previous study [24]. The microtweezers consist of two arms connected by a flexible plate, which is displaced by a piezoelectric bimorph actuator. A microcantilever that functions as a force sensitive tip was attached to each of the two arms. When the tweezers compress a sample, the bending of the tweezer tips and the indentation of the sample are measured by tracking microscopic images. The measured displacements and the known stiffness of the cantilever provide the information needed to find the sample stiffness. The main advantage of our system is that the tweezer tips do not require any active force sensing elements and their bending is simply monitored through microscopic observation. The force sensing tips can easily be changed to match experimental conditions or target objects. In our previous study, we used photolithography to fabricate force sensing tips made of a photopolymer SU-8 (MicroChem, Woburn, MA, USA). However, SU-8 is not approved by the Food and Drug Administration (FDA) as a biocompatible material [25], and it may not be widely acceptable to study the growth of live cells or tissues. In this study, we used a precision-cutting machine to cut a polydimethylsiloxane (PDMS) film into the shape of the force sensing tips. PDMS is an FDA approved, commonly-used material for biological and biomedical applications because of its advantageous properties, including biocompatibility and easy and low-cost fabrication.

We demonstrated the efficacy of our biocompatible force sensitive cantilevers by studying the growth of zebrafish embryos. Zebrafish (*Danio rerio*) are one of the most popular vertebrate animal models in biomedical studies because they are easy to keep and breed, they grow at a fast rate compared to other vertebrate animal models (several days), and their transparent body at the embryonic and larval periods allows researchers to observe their internal structure [26,27]. The zebrafish embryo is thus an excellent platform to study the development and formation of functional tissues and organs in vertebrates. Zebrafish development is traditionally divided into several periods from the one-cell stage to 72 h post-fertilization (hpf), with distinct and well-characterized morphological structures forming in each [28]. During the segmentation period (10–24 hpf), sequential groups of mesodermal cells undergo a striking mesenchymal-to-epithelial transition about every 30 min as they form the somites, in which the dermis, vertebrae, and skeletal muscle begin to differentiate [29]. We hypothesized that significant stiffness changes occur during the segmentation period and we therefore measured the stiffness of the embryos hourly for 10 h to observe changes over this time.

## 2. Materials and Methods

### 2.1. Design of the System

#### 2.1.1. Microtweezers

The microtweezers were comprised of two tweezer arms, each with a cantilever fixed to the ends as the force sensing tip (Figure 1a). The tweezer arms were connected to each other by a flexible plate spring. The tweezer arms and the flexible plate spring were designed using SolidWorks®, and the entire body was 3D printed through selective laser sintering (SLS) using nylon powder (Shapeways, New York, NY, USA). A single bimorph piezo actuator (Steminc, Miami, FL, USA) was set between the moving arm and the tweezer body. When a voltage was applied, the piezo actuator bent and pushed the circular fulcrum of the moving arm, rotating the moving arm about the center of the plate spring. The cantilevers were attached to the tweezer arms using cantilever holders (Figure 1b). The cantilever holders were milled using a monoFab SRM-20 Compact Milling Machine (Roland DGA Corporation, Irvine, CA, USA), and the adjustable holders were 3D printed through stereolithography of ultraviolet (UV) curable acrylic polymer (Shapeways, New York, NY, USA) which can print out structures at a higher resolution than nylon powder.

#### 2.1.2. Principle of Force Sensing

A sample was placed between the two cantilevers of the microtweezers. When the cantilevers compressed the sample, the sample was deformed, and the cantilevers were bent by the applied forces. From Hook’s law, the forces applied in the microtweezer system were described as the following:(1)$$\begin{array}{c} \{\begin{array}{c} F_{1}\cos\theta_{1}=k_{c1}d_{c1}\\ F_{2}\cos\theta_{2}=k_{c2}d_{c2} \end{array}, \end{array}$$
where the numbers 1 and 2 indicate the cantilever on the left and the right respectively, $F_{1}$ and $F_{2}$ are the applied forces, $\theta_{1}$ and $\theta_{2}$ are the angles between the cantilevers and the tangent lines of the sample (Figure 2a), $k_{c1}$ and $k_{c2}$ are the spring constant of the cantilevers, and $d_{c1}$ and $d_{c2}$ are the displacement of the cantilevers (Figure 2b). Biological tissues are non-uniform composite materials which can be modeled as an assembly of multiple segments, as will be discussed in the results section. 

It is practical to model the embryo as a simple spring because it indicates a clear force-displacement relationship and allows us to design cantilevers that better match the sample stiffness.

When we assumed that the stiffness was uniform along the sample and the applied forces at the two cantilever sides were balanced, we could use the resulting equation $F_{1}=F_{2}$ to obtain the following relationship between the forces applied by the cantilevers and the sample indentation:(2)$$\begin{array}{c} F=k_{s}d_{s1}=k_{s}d_{s2}. \end{array}$$
where $k_{s}$ was the spring constant of the sample on each side and $d_{s1}$ and $d_{s2}$ were the sample indentations on the left and the right, respectively (Figure 2c). In our study, we measured the total sample indentation $D_{s}=d_{s1}+d_{s2}$ and the cantilever bending of the fixed arm $d_{c1}$. From these measurements, sample stiffness $k_{s}$ could be calculated by the following equations:(3)$$\begin{array}{c} k_{s}D_{s}=2F=\frac{2k_{c1}d_{c1}}{\cos\theta_{1}}=\frac{2k_{c2}d_{c2}}{\cos\theta_{2}}, \end{array}$$
(4)$$\begin{array}{c} k_{s}=\frac{2k_{c1}d_{c1}}{D_{s}\cos\theta_{1}}=\frac{2k_{c2}d_{c2}}{D_{s}\cos\theta_{2}}. \end{array}$$

#### 2.1.3. Stiffness Analysis Using Pattern Matching and Tracking

In order to measure cantilever bending and sample indentation, we used pattern matching and tracking of the optical images using a custom MATLAB program. While a sample was compressed by the microtweezers with N steps, the sample images of each step were taken by a charge-coupled device (CCD) camera. An image tile of 50×50 pixels was chosen at the edge of the cantilevers from the first image, and a scan area of 100×100 pixels was searched in the second image by the pattern-matching algorithm to find the best matching area of the image tile in the first image. In the algorithm, the dot product of the normalized target vector (the chosen image tile, 50×50=2500 elements) and a normalized subset vector (50×50=2500 elements) of the scan area was calculated as the subset area. The subset vector swept the scan area and, when it gave the maximum dot product with the target vector, it was defined as the best matched area in the second image. Once the best matched area was defined in the second image, it was updated as the new target vector and the scan area in the third image was searched. This process was repeated for N steps, and the movement of the target image tile was calculated in pixels. In this experiment, we measured the displacement of the cantilevers and sample indentations in pixels and converted the measurements to millimeters.

### 2.2. Cantilever

#### 2.2.1. Cantilever Fabrication

The cantilevers were fabricated from a thin film of Polydimethylsiloxane (PDMS). First, a Sylgard 184 Silicone Elastomer base and a curing agent (Dow Corning, Midland, MI, USA) were mixed at a weight ratio of 8:1. We added more curing agent than the typical mixing ratio of 10:1 because stiffer PDMS retained better shapes when cut into small pieces. The PDMS mixture was spin-coated on a glass slide at a speed of 500 rpm at an acceleration of 300 rpm/s for 60 s. It was then cured at 120 °C for 1 h. The fabricated PDMS film with a typical thickness of about 180 μm was cut to cantilevers of length 4 mm and width 300 μm by using a Silver Bullet Cutter (Silver Bullet Cutters, Apple Valley, MN, USA). The cantilevers were attached to the cantilever holders by using a drop of PDMS mixture as a glue.

#### 2.2.2. Cantilever Calibration

The dimensions of the cantilevers were designed so that the cantilevers would be sufficiently soft for stiffness analysis of zebrafish embryos. The spring constant of a cantilever is given by the following equation:(5)$$\begin{array}{c} k=\frac{3EI}{L^{3}}, \end{array}$$
where E is Young’s modulus of the cantilever, I is the second moment of area, and L is the cantilever length. For a rectangular cantilever, the second moment of area is given as $I=\frac{WT^{3}}{12}$ with the cantilever width W and thickness T. Equation (5) can then be written as:(6)$$\begin{array}{c} k=\frac{EWT^{3}}{4L^{3}}. \end{array}$$

According to the literature, the typical Young’s modulus of PDMS, with a mixing ratio of 10:1, is around several hundred kPa to several MPa, and it depends upon various factors, such as curing temperature, curing time, and so forth [30,31,32,33]. Therefore, cantilever calibration was necessary to know the actual spring constant of the fabricated cantilevers. In the calibration, a cantilever made of Polyethylene terephthalate (PET) was used as a reference cantilever. The dimensions of the reference cantilever were L×W×T=20 mm×1 mm×0.13 mm. First, the spring constant of the reference cantilever was measured by using a load cell, rated for 20 gf. The load cell was fixed to a stepper motor and pushed the tip of the reference cantilever while it moved down in 10 steps with about 0.4 to 0.5 mm per step. The applied force was measured by the load cell and the deflection of the cantilever was observed by a CCD camera as it was being bent. After obtaining the spring constant of the reference cantilever, the spring constant of the PDMS cantilevers were obtained in a similar way using the reference cantilever. The PDMS cantilever was fixed on a stepper motor and pushed the reference cantilever tip-to-tip while it moved down in 20 steps with about 0.07 mm each step. The bending distances $\delta_{ref}$ and $\delta_{c}$ of the reference and the PDMS cantilevers, respectively, were observed by a 1288×964 pixel CCD camera (FLIR Systems, Nashua, NH, USA). The force applied to the PDMS cantilevers were calculated from the spring constant and the displacement of the reference cantilever, providing the spring constant of the PDMS cantilevers. Using the ratio of $\delta_{ref}$ over $\delta_{c}$ and spring constant of the reference cantilever $k_{ref}$, the cantilever stiffness $k_{c}$ could be found as $k_{ref}\cdot \left(\frac{\delta_{ref}}{\delta_{c}}\right)$.

### 2.3. Experimental Setup

Figure 3a shows our experimental setup. A microscope composed of a 1288×964 pixel CCD camera (Point Gray) and an M PLAN APO 5X/0.14 objective lens (MITUTOYO, Kawasaki, Japan) were used. An Arduino® Uno board was used as the serial communication interface for microtweezer opening/closing control. The input voltage of −45 V to +45 V was supplied from the Arduino board through a high voltage amplifier to the piezo electric actuator, according to the commands from the MATLAB program. In the experiment, 30 steps of input voltage were applied to the piezo actuator to close the microtweezers and apply indentation to embryos. Figure 3b shows a typical plot of the distance between the two cantilevers for 30 steps of input voltage.

### 2.4. Preparation of Zebrafish Embryos

Zebrafish embryos at the beginning of the segmentation period were selected and manually dechorionated before the experiment. During the experiment, the dechorionated embryos were kept in a 35 mm tissue-culture treated dish (Celltreat, Pepperell, MA, USA) filled with the embryo media (13.7 mM NaCl, 0.5 mM KCl, 1.3 mM CaCl_2_, 1 mM MgSO_4_, 4.2 mM NaHCO_3,_ 0.07 mM sodium/potassium phosphate buffer, and pH 7.2). The conventional zebrafish developmental staging series was based on an incubation temperature of 28.5 ℃, with increases or decreases in temperature of a few degrees speeding or slowing development, respectively, without detrimental effect [28]. The temperature during the experiment was approximately 25–27 ℃. We used two zebrafish embryos (referred to as Embryos 1 and 2) for the stiffness analysis. Figure 4 shows growth of Embryo 1 during the 9 h experiment, in which the embryo developed from approximately the 3-somite stage to the 20-somite stage.

## 3. Results

### 3.1. Cantilever Calibration

The obtained spring constant of the reference cantilever was $k_{ref}=0.151$ N/m. From the dimensions of the cantilever (L×W×T=20 mm×1 mm×0.13 mm) and Equation (6), the elastic modulus of the reference cantilever material is approximately 2.20 GPa, which is within the observed range of the elastic modulus of PET of 2–2.7 GPa [34]. Figure 5 shows the force measurement corresponding to the bending distance of the PDMS cantilever we used in the study. The equation of the linear regression is $y=1.48\times 10^{-2}x$, where x is the bending distance of the PDMS cantilever (mm) and y is the force (mN). From the slope of the linear regression, the calibrated stiffness of the PDMS cantilever was $1.48\times 10^{-2}$ [N/m]. To evaluate manufacturing an error of PDMS cantilevers, we made 10 identical PDMS cantilevers and measured the dimensions, the spring constants, and the Young’s moduli of them. Table 1 shows the average and the standard deviations of the measurements among 10 PDMS cantilevers. The average Young’s modulus of the 10 PDMS cantilevers was estimated to be 1.70±0.77 MPa, which is within the range of reference values of 1.3–2.5 MPa reported in [32,33].

### 3.2. Stiffness Analysis of Zebrafish Embryo

The displacement of the cantilever at the fixed arm: $d_{c1}$ and the total sample indentation: $D_{s}$ were obtained from pattern matching, and the cantilever angle $\theta_{1}$ was measured by ImageJ. We calculated the stiffness of zebrafish embryos from Equation (4). Figure 6 shows the average stiffness of Embryos 1 and 2 at each experimental time point. The deviation of the determination was calculated as $R^{2}=0.718$, which was comparable to values reported in studies of biosample stiffness measurements [35,36].

We also measured strains of the zebrafish body and the yolk for each embryo. Figure 7 shows the images of Embryo 2 at T = 0 (~ 3-somite stage) and 9 (18 to 20-somite stage) that was compressed by the cantilevers. One can see that the deformation in the body is much greater than in the yolk in (a), while the deformation in (b) became less visible. Figure 8 shows the average strains of the bodies and yolks of Embryos 1 and 2. We measured the distances along the body and the yolk using ImageJ to calculate the strains. The measurements show that the zebrafish body was softer than the yolk at the early stages of the segmentation period; the body became stiffer than the yolk at around T = 6–8 h. The strain on the yolk was found to remain similar throughout the measurements, suggesting that the elasticity of the yolk part does not change as much as that of the body. 

### 3.3. Young’s Modulus Estimation Using a Finite Element Analysis

The Young’s moduli of the zebrafish embryos were estimated using the finite element analysis software COMSOL Multiphysics (version 5.2). In the analysis, the COMSOL solid mechanics (stationary) module was used, and a zebrafish embryo model was designed using SolidWorks®. The model was simplified to a curved tube component and a sphere component, presenting the body and the yolk, respectively (Figure 9b). The key dimensions of the model were approximated from the average values measured from the images of zebrafish embryos. In our model, the diameter of the yolk was set to 0.6 mm, and the heights of the head, center body, and the tail from the yolk contact area were set to 0.17 mm, 0.16 mm, and 0.14 mm respectively. The cantilever contact area on the right and left sides of the embryos were also defined according to the measurement from their images. The vertical lengths of the cantilever contact areas at the body side and the yolk side were set to 0.2 mm and 0.35 mm respectively. For the finite element analysis, a finer free tetrahedron mesh was used. A fixed constraint was applied to the left contact area that is dorsal and centered on the anterior-posterior (AP) axis, and a Boundary Load was applied to the left contact area that is ventral and centered on the AP axis. The values of the boundary load were found from the measurements of the cantilever bending and Equation (3). The image of zebrafish embryos at T = 8 h was used for this analysis. We chose this time point because one can observe well-developed zebrafish bodies then and the stiffness changes can be attributed to its structural formation. The Young’s moduli of the zebrafish body and yolk were iteratively adjusted and optimized through the hill-climbing method to match the measured strain values at the body and the yolk. The estimated Young’s modulus of the zebrafish body was around 170 Pa and the estimated Young’s modulus of the yolk was around 48 Pa. The estimated values were convincing as they were close to the Young’s moduli of human epithelial cells (about 50–100 Pa), measured by magnetic twisting cytometry and optical tweezers, and the cancerous human epithelial cells (about 200–400 Pa) were measured by the scanning force microscopy and AFM reported in other literature [16,37,38].

## 4. Discussion and Conclusions

The results of our stiffness analysis show a gradual increase in stiffness of zebrafish embryos over time. The results of the strain measurements indicate that the stiffness of the zebrafish body at the onset of the segmentation period rapidly increases within several hours, while that of the yolk remains similar. From the finite element method (FEM) analysis, the elastic moduli of the body and yolk at T=8 h were estimated to be 170 Pa and 48 Pa, respectively.

Because of the contained liquid, live cells and tissues are viscoelastic materials that have both elastic and viscous properties [24]. However, when the process of compression is slow enough, an assumption can be made that the tissue deformation is quasi-static. Our prior work has shown that compression with intervals of 1 s for the total of ~30 compression steps is slow enough so that viscosity is negligible [24].

Following the experiments, most zebrafish embryos survived and became healthy zebrafish larvae with no apparent defect, suggesting that our microtweezer system does not impede their development and is suitable for long term experiments.

A critical aspect of the stiffness analysis of zebrafish embryos was the location at which tweezer indentation applied. We set the cantilever of the fixed arm to the dorsal and the center of the AP axis of zebrafish embryos and the cantilever of the moving arm to the ventral and the center of the AP axis of them in order to avoid slipping of their bodies from the cantilever surface. However, it was still challenging to measure stiffness of embryos beyond the segmentation period as their structure become more complex and the embryos moved in response to physical stimuli. Future work will include the development of a method to firmly fix their posture during the measurements without inhibiting morphogenesis.

In conclusion, we demonstrated the measurement of stiffness changes during the growth of zebrafish embryos. The results provided good indications of the structural changes in the body during the segmentation period. The results of the COMSOL analysis also contributed to estimations of the Young’s modulus of the zebrafish body and yolk at later stages in the segmentation period. The cantilevers made of PDMS, which is an elastic and biocompatible material, did not cause any apparent negative effects on the growth of zebrafish embryos.

## Figures and Tables

**Figure 1 sensors-19-01506-f001:**
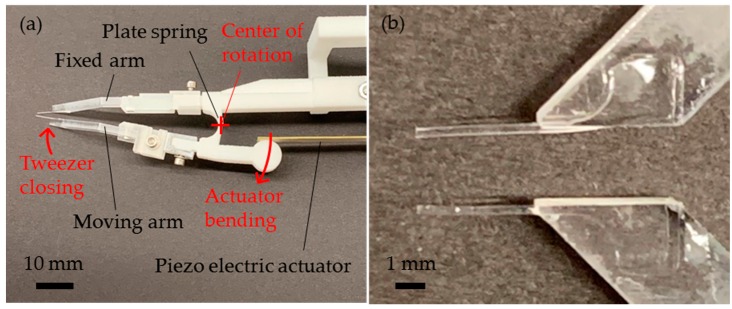
Photographs of the microtweezers. (**a**) The microtweezer system, (**b**) polydimethylsiloxane (PDMS) cantilevers attached on the acrylic cantilever holders.

**Figure 2 sensors-19-01506-f002:**
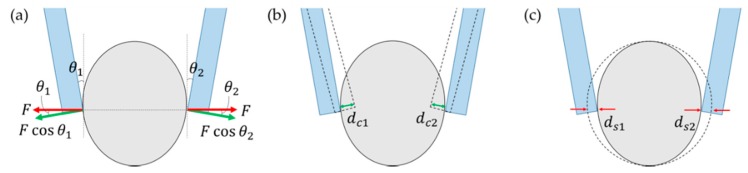
Force sensing by the microtweezers. (**a**) Force diagram. (**b**) Deflections of the cantilevers. (**c**) Sample indentations.

**Figure 3 sensors-19-01506-f003:**
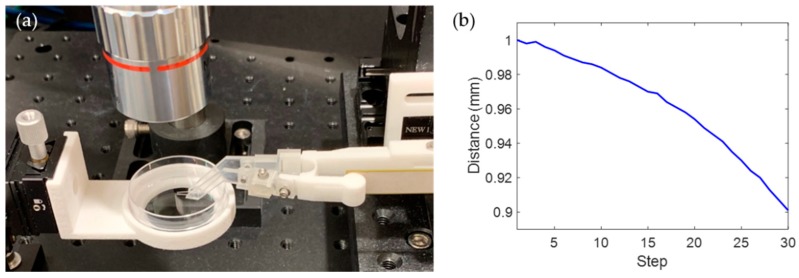
(**a**) Picture of the experimental setup, (**b**) plot of the distance between the two cantilevers for 30 steps.

**Figure 4 sensors-19-01506-f004:**
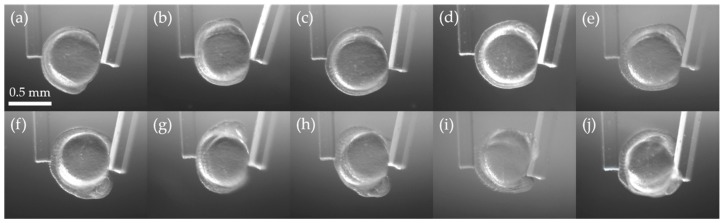
Growth of Embryo 1 during the experiment. (**a–j**) are pictures at the experimental time T = 0, 1, …, 9 h (approximately 3 to 20-somite stages).

**Figure 5 sensors-19-01506-f005:**
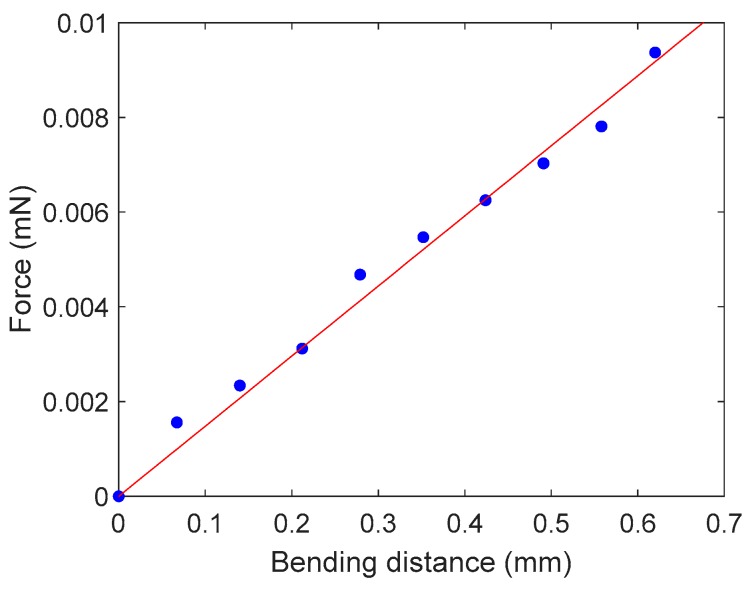
Force measurement of the PDMS cantilever at the fixed arm side. The red line is the linear regression: $y=1.48\times 10^{-2}x$ ($R^{2}=0.987$).

**Figure 6 sensors-19-01506-f006:**
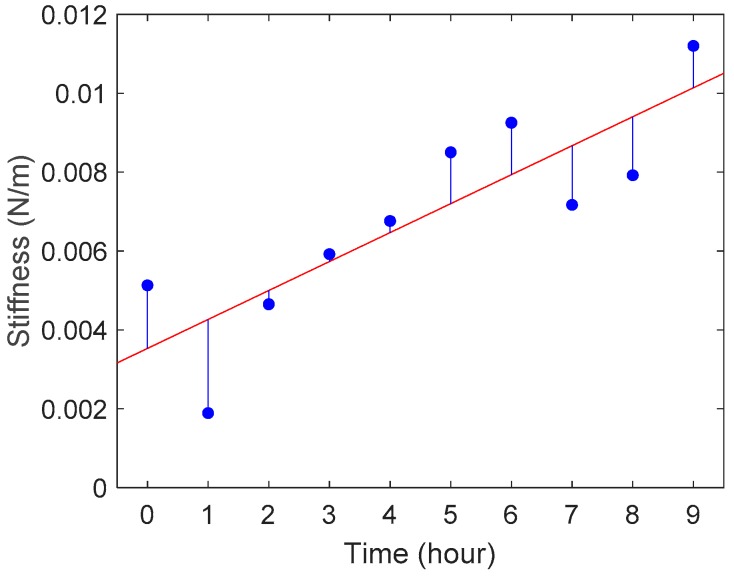
Average stiffness changes during the growth of Embryo 1 and Embryo 2. The red line is the linear regression: $y=7.34\times 10^{-4}x+3.53\times 10^{-3}\;\left(R^{2}=0.718\right)$.

**Figure 7 sensors-19-01506-f007:**
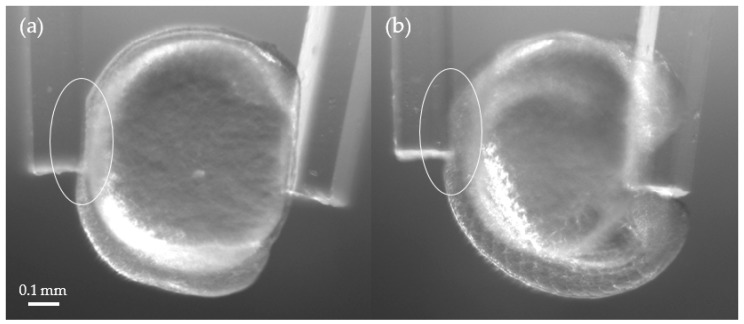
(**a**) Compression of Embryo 2 at T = 0 (~ 3-somite stage). The body showed a larger deformation than the yolk. (**b**) Compression on Embryo 2 at 9 (18 to 20-somite stage). The deformation of the body was significantly reduced.

**Figure 8 sensors-19-01506-f008:**
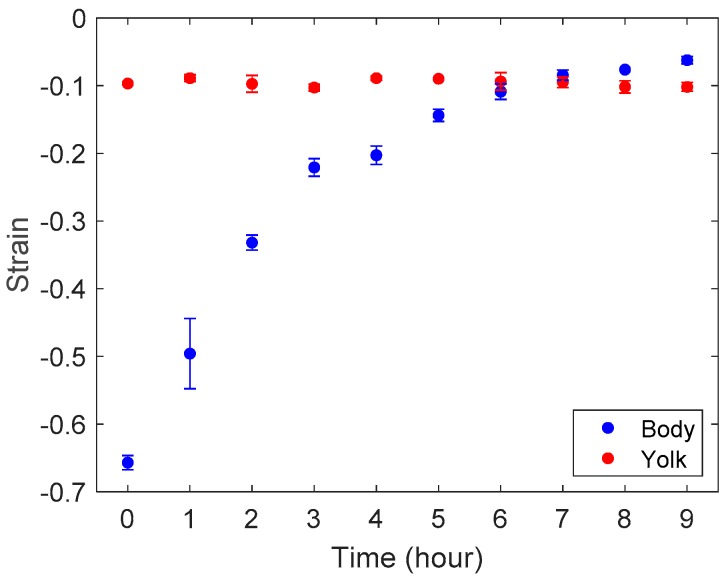
Average strains of the zebrafish body and the yolk at each time point.

**Figure 9 sensors-19-01506-f009:**
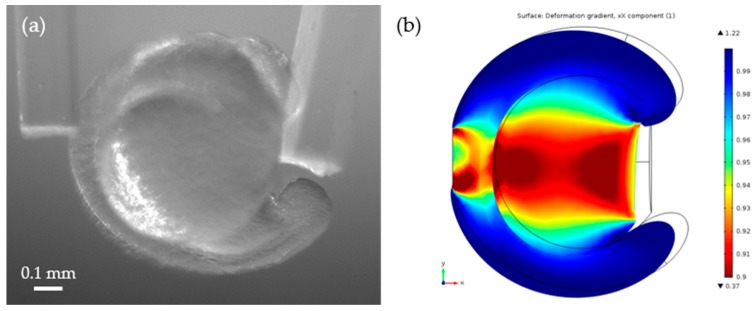
(**a**) Embryo 2 at T = 8 h, (**b**) strain analysis of the zebrafish embryo using COMSOL Multiphysics. The curved tube is the body of the zebrafish embryo and the round part is the yolk.

**Table 1 sensors-19-01506-t001:** Measured dimensions of PDMS cantilevers (N = 10).

	Length (mm)	Width (mm)	Thickness (mm)
Average	3.47	0.261	0.183
Standard deviation	0.16	0.015	0.044

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
