# Peer review of "Biocompatible Cantilevers for Mechanical Characterization of Zebrafish Embryos using Image Analysis"

_sensors, 2019, doi:10.3390/s19071506_

Round 1

Reviewer 1 Report

The authors propose a method for estimating stiffness of Zebrafish embryos using a tailor-made device. The results are quite interesting, but it would be worth clarifying some aspect of the method they have used for obtaining these results. In particular equations (1-4) used to estimate stiffness are based on the assumption that sample stiffness is uniform, but experimental results clearly indicate that this is not the case (the yolk is significantly softer than the body, for example. How reliable are these stiffness estimates then? 

It would be important to quantify variability in the measurement. For example, the two plots in Figure 5 are rather different...is this an indication of unreliability in stiffness measurement (see above), measurement error or inter-embryo variability?

Methodology discussed in section 3.3 is not very clear either. How did the author model the embryos? Are they matching the morphology of the yolk/body based on the images, or are they using a standard template for body and yolk? How was the presence of fluid in the yolk taken into account?

In summary, I believe the paper contains interesting bits, but methodology needs to be explained more rigorously in order to generate the required trust/confidence in the reader.

Author Response

We thank the reviewer for the detailed comments. Here we would like to address the suggestions given in the review.

(Comment 1) Equations (1-4) used to estimate stiffness are based on the assumption that sample stiffness is uniform, but experimental results clearly indicate that this is not the case (the yolk is significantly softer than the body, for example.

(Response 1) We have added the description to clarify the reason why we introduced the assumption of uniformity:

“Biological tissues are non-uniform composite materials which can be modeled as an assembly of multiple segments as will be discussed in the results section. However, it is practical to model the embryo as a simple spring because it indicates a clear force-displacement relationship and allows us to design cantilevers that better-match the sample stiffness.”

(Comment 2) How reliable are these stiffness estimates? It would be important to quantify variability in the measurement. For example, the two plots in Figure 5 are rather different...is this an indication of unreliability in stiffness measurement (see above), measurement error or inter-embryo variability?

(Response 2) As suggested, we have quantified the variability using the average of the measurements. Deviation found in our study is comparable to the ones reported in literature. Because biological tissues are irregularly-shaped samples, biosample stiffness measurements, including studies by other groups [35,36], tend to show some deviations. Please see the revised Figure (now it is Figure 6) and the newly added description to discuss the viability.

“The deviation of the determination was calculated as R2 = 0.718, which was comparable to values reported in studies of biosample stiffness measurements [35,36].”

(Comment 3) Methodology discussed in section 3.3 is not very clear either. How did the author model the embryos? Are they matching the morphology of the yolk/body based on the images, or are they using a standard template for body and yolk? How was the presence of fluid in the yolk taken into account?

(Response 3) We have made the simplified model of the embryo based on the microscopic images. We have added the following description:

“The model was simplified to a curved tube component and a sphere component, presenting the body and the yolk, respectively (Figure 9b). The key dimensions of the model were approximated from the average values measured from the images of zebrafish embryos. In our model, the diameter of yolk was set to 0.6 mm, and the heights of the head, center body, and the tail from the yolk contact area were set to 0.17 mm, 0.16 mm, and 0.14 mm respectively.”

We have also added a discussion on the effect of the liquid contained in the tissue.

“Because of the contained liquid, live cells and tissues are viscoelastic materials that have both elastic and viscous properties [24]. However, when the process of compression is slow enough, an assumption can be made that the tissue deformation is quasi-static. Our prior work has shown that compression with intervals of 1 s for the total of ~30 compression steps is slow enough so that viscosity is negligible [24].”

(Comment 4) In summary, I believe the paper contains interesting bits, but methodology needs to be explained more rigorously in order to generate the required trust/confidence in the reader.

(Response 4) Thank you for the encouraging comment.

Reviewer 2 Report

The manuscript describes the force sensing scheme based on the static deflection of a set of cantilevers and mechanical characterization of zebrafish embryos. Since the working principle and the demonstration of the technique has been previously reported by the authors, the merit of the manuscript would be the fabrication of PDMS-based cantilever, which is biocompatible, and the characterization of zebrafish embryos at various growth stages, as the authors claim. The work presented would be of interests to the researchers working on both mechanical sensor systems and biomechanics. However, I recommend the authors to clarify some of the issues prior to publication.

What differentiates this manuscript from the previous work would be the use of the PDMS-based cantilever. In addition, the spring constant of the PDMS-cantilever is critical to later characterization of the embryos. Though, the spring constant of the cantilever is only presented with a single fitting function without indicating what x and y in the function mean, while the description of the reference cantilever is given in more details. It would be better to provide rather detailed information on the PDMS cantilever, such as a measurement graph and elastic modulus as well as the fitting function, for the benefit of the readers. Is the elastic modulus in the range of value for PDMS, reported in other literatures?

What is the range of errors in this work? For example, if one looks at Fig. 5, two embryos show small differences in stiffness at each growth time. Are these from the experimental uncertainties or embryo-dependent deviations? To clarify this, the error bars in the graph and short comment on the experimental uncertainty will be appreciated.

Somewhat relating to the above question 2, it is not clear what the purpose of presenting separate experimental data from two embryos 1 and 2 is in the main text. It would be helpful to remark on the reason of performing two independent measurements and results regarding the deviation between them.

Are the obtained mechanical characteristics of the embryos such as Young’s modulus comparable to those observed from other works, if there is any, or to those of similar cell structures? This will also confirm the validity of the experimental approach.

Author Response

We thank the reviewer for the detailed comments. Here we address the comments given in the review.

(Comment 1) The work presented would be of interests to the researchers working on both mechanical sensor systems and biomechanics. However, I recommend the authors to clarify some of the issues prior to publication.
(Response 1) Thank you for the encouraging comment.

(Comment 2) What differentiates this manuscript from the previous work would be the use of the PDMS-based cantilever. In addition, the spring constant of the PDMS-cantilever is critical to later characterization of the embryos. Though, the spring constant of the cantilever is only presented with a single fitting function without indicating what x and y in the function mean, while the description of the reference cantilever is given in more details. It would be better to provide rather detailed information on the PDMS cantilever, such as a measurement graph and elastic modulus as well as the fitting function, for the benefit of the readers. Is the elastic modulus in the range of value for PDMS, reported in other literatures?

(Response 2) We have made the following revisions:

We have added the calibration graph (see Figure 5) for the cantilever sensor we used in the testing.

We measured the dimensions (L x W x T) for 10 PDMS cantilevers and summarized the measurement in Table 1.

We added the description: "The average Young’s modulus of the 10 PDMS cantilevers was estimated to be 1.70±0.77 MPa, which is within the range of reference values of 1.3 – 2.5 MPa reported in [32,33]".

(Comment 3) What is the range of errors in this work? For example, if one looks at Fig. 5, two embryos show small differences in stiffness at each growth time. Are these from the experimental uncertainties or embryo-dependent deviations? To clarify this, the error bars in the graph and short comment on the experimental uncertainty will be appreciated.

(Response 3) Deviation found in our study is comparable to the ones reported in literature. Because biological tissues are non-uniform, irregularly-shaped samples, biosample stiffness measurements, including studies by other groups [35,36], tend to show some deviations. In order to clarify this, we have quantified the deviations using the average of the measurements and added error bars in the graph. Please see the revised Figure (now it is Figure 6) and the newly added description to discuss the viability.

“The deviation of the determination was calculated as R2 = 0.718, which was comparable to values reported in studies of biosample stiffness measurements [35,36].”

(Comment 4) Somewhat relating to the above question 2, it is not clear what the purpose of presenting separate experimental data from two embryos 1 and 2 is in the main text. It would be helpful to remark on the reason of performing two independent measurements and results regarding the deviation between them.
(Response 4) As suggested, we have calculated the average of the measurement and combined the graph. Please see the revised Figure (now it is Figure 8). Error bars have been added to indicate that the two measurements are showing a very similar trend.

(Comment 5) Are the obtained mechanical characteristics of the embryos such as Young’s modulus comparable to those observed from other works, if there is any, or to those of similar cell structures? This will also confirm the validity of the experimental approach.

(Response 5) To the best of our knowledge, our study is the first to measure the stiffness of zebrafish embryos. However, the result is comparable to the values reported for other types of similar biosamples. We have added the new description:

“The estimated values were convincing as they were close to the Young’s moduli of  human epithelial cells (about 50 – 100 Pa) measured by magnetic twisting cytometry and optical tweezers, and cancerous human epithelial cells (about 200 – 400 Pa) measured by scanning force microscopy and AFM reported in other literature [37-39]”

Round 2

Reviewer 2 Report

In the revised manuscript, the authors have further discussed and clarified the concerns raised by the referee, which guarantees the publication in Sensors.